

# Adaptation and validation of the Spanish version of the Being a Mother scale

Anna Riera-Martín[1,2], Antonio Oliver-Roig[3], Susana Cormenzana[1], Miguel Richart-Martínez[3] and Ana Martínez-Pampliega[1]

[1] Department of Psychology, University of Deusto, Bilbao, Spain
[2] Biogipuzkoa Health Research Institute, Donostia-San Sebastián, Guipúzcoa, Spain
[3] Department of Nursing, University of Alicante, Alicante, Spain

## ABSTRACT

**Background**. Becoming a mother is a very important process because of the impact it can have on women and their families. Currently, there is no validated questionnaire that evaluates the process of becoming a mother in the Spanish population. Moreover, no consistent results have been obtained to identify significant differences between primiparous and multiparous mothers.

**Aim**. (1) Linguistic and metric validation of the Being a Mother scale (BaM-13) in the Spanish population, (2) analysis of possible differences between primiparous and multiparous mothers' experience of motherhood.

**Methods**. Instrumental design. In 2016–2017, a sample of 579 mothers with children between 6 and 11 months of age completed the Spanish version of BaM-13. The instrument was translated using forward and back translation. Construct validity, internal consistency, and criterion validity were empirically analyzed.

**Results**. Factorial analyses showed that the scale presented two adequate factors. Internal consistency of the global scale ($\alpha = 0.818$, $\omega = 0.861$), the Postnatal bonding factor ($\alpha = 0.773$, $\omega = 0.784$), and the Adult's experience factor ($\alpha = 0.710$, $\omega = 0.721$) was adequate. Significant associations were found with postpartum depression ($r = 0.560$), parental competence ($r = -0.584$) and postnatal bonding ($r = -0.327$). In terms of parity, primiparous mothers have greater difficulty in postnatal bonding, compared to multiparous mothers ($p = 0.006$).

**Conclusions**. The Spanish version of the BaM-13 scale is valid for measuring mothers' experience of motherhood in a wide range of domains. The findings of the study show the importance of considering parity in the experience of becoming a mother, highlighting the approach to postnatal bonding in primiparous mothers. Additionally, we underline that it should not be assumed that multiparous mothers experience fewer difficulties in their motherhood process.

Corresponding author
Miguel Richart-Martínez,
m.richart@ua.es

## INTRODUCTION

Becoming a mother is one of the most important vital processes that occur in the perinatal period (*Fowles & Horowitz, 2006*). This period is a stage of high vulnerability for women's mental health, and the process of becoming a mother is particularly relevant because of

the negative consequences that could be generated in the family system: in the mother, the couple relationship, and the baby. As for the mother's consequences, increases in psychological disorders, such as postpartum depression, anxiety disorders, obsessive-compulsive disorders, posttraumatic stress disorders, and psychotic episodes, among others, have been observed (*Gavin et al., 2005*; *Munk-Olsen et al., 2006*). Regarding the consequences in the couple relationship, greater marital conflict, less satisfaction with the quality of the relationship, lower sexual satisfaction and intimacy, as well as less joint leisure time have been observed (*Claxton & Perry-Jenkins, 2008*; *Doss et al., 2009*). Finally, all these variables can have a negative impact on the cognitive, emotional, and behavioral development of the baby (*Stein et al., 2014*).

The scientific literature presents different theoretical approaches to the concept of becoming a mother. Firstly, deductive approaches, generated by quantitative research, study the patterns of maternal behavior during the transition to motherhood, as well as variables that influence this process (*Mercer, 2010*). In this line, we highlight the Maternal Role Attainment (MRA) theory proposed by *Rubin (1984)* and developed by *Mercer (1985)*. Secondly, the inductive approaches, generated by qualitative research, describe the process from the experience of the women and their socially constructed context (*Nelson, 2003*). Finally, the third approach proposes to integrate these two approaches to establish a common theoretical framework of the concept of becoming a mother. In this line, the Become a Mother (BAM) theory stands out, inscribed within the theories of transition and proposed by Mercer (*Mercer, 2004*; *Mercer, 2010*). It redefines the MRA theory, incorporating the findings found in qualitative research and the contributions of *McBride & Shore (2001)* to provide an overview of the process. The main constructs of the BAM theory are parental competence and postnatal bonding (*Mercer, 2010*). Likewise, maternal mental health and having or not having previous children (parity) are variables that can affect the process (*Mercer, 2010*). Our study falls within this theoretical approach, and becoming a mother, the experience of motherhood, or the transition to motherhood are the terms that will be adopted.

Although there is a consensus in the literature on how mental health is affected during the process of becoming a mother, the same cannot be stated about parity, as this variable has several inconsistencies: the first, of a more theoretical nature, deals with the inclusion or exclusion of multiparity in the process of becoming a mother; and the second is the disparity in the results of studies comparing mental health in primiparous and multiparous mothers. Concerning the first inconsistency, the scientific literature tends to relate the concept of transition to motherhood or becoming a mother with primiparous mothers (*Gameiro, Moura-Ramos & Canavarro, 2009*). However, several authors point out that the birth of a child, regardless of whether it is the first or a later child, requires a reorganization in maternal identity and behavior, so multiparous mothers must also be incorporated into the concept of becoming a mother or transition to motherhood (*Chapman & Hart, 2017*; *Mercer, 2004*). As for the second inconsistency, compared with the studies that determine that primiparous mothers have more difficulties in the process of becoming a mother (*Martínez-Galiano et al., 2019*), other research concludes that multiparous mothers show higher rates of stress and mental health difficulties (*Canário & Figueiredo, 2017*). Some

studies also find no significant differences between the two groups of mothers (*Bassi et al., 2017*; *Osnes et al., 2021*).

In order to detect and prevent difficulties in the process of becoming a mother, it is necessary to have valid and reliable tools to assess the experience of motherhood in its different domains, not just mood. Some of the questionnaires used for this purpose are: the Experience of Motherhood Questionnaire (EMQ) (*Astbury, 1994*), the Being a Mother Scale (BaM-13) (*Matthey, 2011*), the Maternal Experience Scale (MES) (*DiPietro et al., 2015*) and the Barkin Index of Maternal Functioning (BIMF) (*Barkin, Wisner & Wisniewski, 2014*).

The BaM-13 scale (*Matthey, 2011*) was selected for different reasons. On the one hand, despite the lack of explicit adherence to a particular theory, this scale follows the integrative approach proposed by Meleis and Mercer (*Mercer, 2010*), integrating the concepts of qualitative and quantitative research and emphasizing the continuous development of the maternal identity (*Mercer, 2004*; *Mercer, 2010*), evaluating the process of becoming a mother up to four years later. Its trans-theoretical nature and ease of application allow its use in primary, secondary, and tertiary care, so it has been used in multiple studies (*Henshaw et al., 2015*; *Pontoppidan, 2015*). From a psychometric point of view, it has adequate reliability and validity (*Matthey, 2011*). To date, it has been adapted in the United States (*Henshaw et al., 2015*), Denmark (*Pontoppidan, 2015*) and Germany (*Göbel et al., 2020*), but there is no Spanish version. A validated version would be useful to identify and prevent difficulties in the process of becoming a mother, and to advance the identification of risk groups, for example, evaluating the influence of parity.

Therefore, this study was conducted with a dual objective: first, to translate the scale into European-Spanish and test its psychometric properties; and second, to analyze the differences between primiparous and multiparous mothers in the process of becoming a mother.

## MATERIALS & METHODS

### Participants

The design of the study is instrumental, and it is part of a larger cohort study on positive parenthood (the PIPP Project). The study initially included a convenience sample of 579 women, who had been recruited at postpartum discharge from 16 hospitals in eastern Spain and who completed an online form between six and eleven months postpartum. The inclusion criteria of the participants in the study were as follows: to be a mother of a full-term newborn, with a low or medium-risk gestation and delivery according to the classification of the program of mothers' care of the Ministry of Health of the Valencian Community (Spain), and not presenting difficulties to speak and read the Spanish language. The recruitment process is detailed in Fig. 1.

### Variables and instruments
#### Becoming a mother
The BaM-13 scale (*Matthey, 2011*). This questionnaire evaluates the experience of motherhood in the period between the first postpartum weeks and four years. It consists of 13 items and uses a four-point Likert scale ranging from 0 (*No, rarely or never*) to 3 (*Yes,
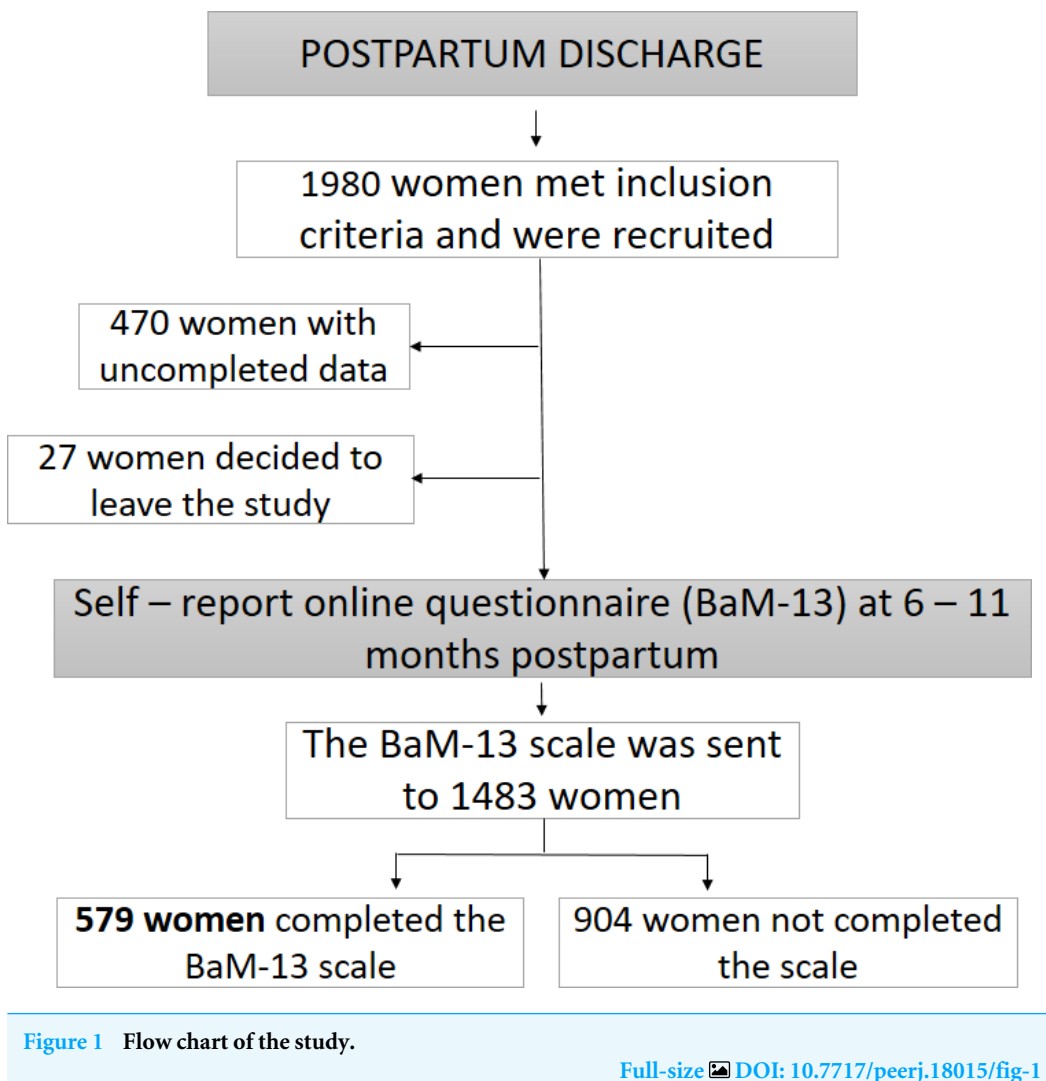

**Figure 1** Flow chart of the study.

*most or all of the time*). Items 1, 4, and 8 and their response options are reverse-written, so their scores will be in the same direction as the remaining items. The total score can range from 0 to 39, with higher scores indicating lower satisfaction with the maternity experience. A score of 9 or more indicates the presence of problems in the maternity experience. The scale groups the items into three factors: the first factor, called Child experience (six items: 3, 9, 10, 11, 12, 13), is defined as the mother's experience with her baby; the second, Adult's experience (five items: 2, 5, 6, 7, 8) values her experience as an adult; the third, Emotional closeness (two items: 1, 4) evaluates the mother's emotional bond with her baby. The original version shows adequate internal consistency (Cronbach alpha of 0.798) and an adequate criterion validity with postpartum depression (Pearson correlation coefficient of 0.64).

### Postpartum depression

The Edinburgh Postnatal Depression Scale (EPDS) (*Cox, Holden & Sagovsky, 1987*): This study used the Spanish version of EPDS (*Garcia-Esteve et al., 2003*). It is a ten-item questionnaire that uses a four-point Likert scale to assess the presence of postpartum depression. The score can range from 0 to 30; higher scores indicate a greater presence of depressive symptomatology. The cut-off point identified to detect the presence of postpartum depression was 10–11 (*Garcia-Esteve et al., 2003*). This version shows adequate internal consistency (Cronbach alpha of 0.79) and a two-factor structure: Sadness and Anxiety.

### Perceived parental competence

The Parenting Sense of Competence (PSOC) (*Johnston & Mash, 1989*): In this study, we used the Spanish version PSOC (*Oltra-Benavent et al., 2019*), which maintains the structure of the original scale. This is a 17-item questionnaire that uses a six-point Likert response scale (1 = *Strongly Disagree*; 6 = *Strongly Agree*). This version shows adequate total internal consistency (Cronbach alpha of 0.89) and factorial structure: Efficacy (0.91) and Satisfaction (0.87).

### Postnatal bond

The Maternal Postnatal Attachment Scale (MPAS) (*Condon & Corkindale, 1998*): This study used the reduced Spanish version of MPAS, called the Postnatal Bond Scale (PBS) (*Riera-Martín et al., 2018*). This is a 15-item questionnaire, rated on a five-point Likert scale that evaluates the postnatal bond of both parents. It maintains the original structure of three factors: Quality of bonding, Absence of hostility, and Pleasure in interaction. The internal consistency of the global scale has shown appropriate values (Cronbach alpha of 0.70), as well as appropriate composite reliability (CR) values in each factor: 0.74 for Quality of bonding, 0.93 for the Absence of Hostility, and 0.83 for Pleasure in interaction.

The authors have permission to use these instruments from the copyright holders.

## Translation procedures

A translation-back-translation process was followed (*Bundgaard & Nisbeth, 2018*): Two bilingual translators made the European-Spanish translation, and two different bilingual translators made the back-translation into English. The translators worked independently and were unaware of the objectives of the study. After this process, a committee of experts examined the two English versions (original and back-translated). Finally, cognitive interviews were performed with ten women to verify the interpretation of the items, the cultural relevance, and the ease of comprehension of the instrument.

## Data collection and ethical considerations

After the participants were recruited, their contact details were recorded, and they were asked to complete a battery of forms. Clinical data about childbirth and early puerperium were obtained through the clinical history at postpartum discharge. At six months postpartum, a battery of online questionnaires was sent to them, with a link containing a unique code for each participant that allowed direct access to the self-reported form,

which could be completed using a Web browser. This process was repeated twice for those participants who had not completed the questionnaire, with a period of 10 days between the two messages. Prior to sending the online questionnaires, we sent reminders *via* mobile phone (SMS).

The study was approved by the Ethical Committee of Clinical Research of the General Direction of Public Health and Higher Center of Research in Public Health (CEIC-DSGSP/CSISP), attached to the Council of Health of the Valencian Community. The participants were informed of the study and gave informed consent by signing a written document. Only the members of the research team had access to the personal data, which were replaced in the forms and databases by an alphanumeric code for each participant, in order to guarantee confidentiality.

## Data analysis

Statistical analyses were performed with the SPSS software program (version 22.0) and the R statistical program, version 3.6.1 for Windows (*R Core Team, 2019*). Categorical variables were represented by absolute frequency and percentage (%). The Kolmogorov–Smirnov (K-S) test was performed to check whether the distribution of the items and the factors were adjusted to the normal distribution. If they were not, the median and interquartile range (IQR) were calculated for the continuous variables. Psychometric properties of the scale were evaluated according to the COSMIN checklist (Consensus-Based Standards for the Selection of Health Status Measurement Instruments), which is used to evaluate the methodological quality of studies on health status measurement instruments (*Mokkink et al., 2010*).

To determine the validity of the construct, an exploratory factorial analysis was performed using principal components analysis with varimax rotation. The adequacy of the data for factor analysis was examined using Bartlett's test of sphericity and the Kaiser-Meyer-Olkin (KMO) measure where a KMO value above 0.8 was considered adequate (*Pett, Lackey & Sullivan, 2003*; *Shrestha, 2021*). To determine the number of factors to be retained, the screeplot of the eigenvalues was reviewed by applying the last-elbow rule between eigenvalues greater than 1, taking into account the interpretability of the factors. A minimum factor loading of 0.4 was used as the criterion for each retained item and a difference of $\geq 0.2$ between the primary loading and any secondary loading for an item; items without this difference were assigned to the factors that made the most theoretical sense (*Stevens, 1992*). Based on the results of the principal components analysis, a confirmatory factorial analysis (CFA) was performed, using the method of diagonally weighted least squares (DWLS) (*Christoffersson, 1977*), with DWLS being an appropriate method for ordinal data nonnormally distributed with a sufficiently large sample size. The model resulting from the principal components analysis was compared with the original structure of the scale, composed of three factors (*Matthey, 2011*) and also with the most parsimonious model: one factor. The following goodness-of-fit indices were analyzed: the Chi-squared ($\chi^2$); the ratio between $\chi^2$ and the degrees of freedom ($\chi^2/df$), with a value less than 3 being optimal (*Bollen, 1989*); the root mean square error of approximation (RMSEA), with its 90% confidence interval, and the standardized root-mean-square

residual (SRMR), considering a value less than 0.08 acceptable (*Hu & Bentler, 1999*); the comparative fit index (CFI), with values greater than 0.90 considered appropriate (*Bentler & Bonett, 1980*), and the Tucker-Lewis index (TLI), with values greater than 0.95 considered appropriate (*Hu & Bentler, 1999*).

To study the internal consistency, Spearman correlation coefficients were calculated to determine the relationships between the dimensions, the Cronbach alpha coefficient ($\alpha$), as well as the omega coefficient ($\omega$) (*McDonald, 1999*), with the recommended values being equal to or greater than 0.70. Additionally, for each item, the mean, standard deviation, skewness index, Spearman correlation coefficient with its factor, and the value of the Cronbach alpha coefficient if the item is removed were calculated.

For the study of the criterion validity, Spearman correlations between the BaM-13 factors and the total score with postpartum depression, postnatal bond, and parental competence were calculated.

The existence of differences in their experience of motherhood between primiparous and multiparous mothers was analyzed through the nonparametric Wilcoxon test. Values of $p < 0.05$ were considered statistically significant.

# RESULTS

## Characteristics of participants

The sample consisted of 579 women with an average age of 34.2 years, all with children between 6 and 8 months of age. Most of the participants were Spaniards (95.5%), married or common law couples (79.6%), lived with their partner (96.2%), had a university-level education (55.4%), and an income of between 12,000 and 45,000 euros per year (67.9%). In terms of parity, 56.8% of the sample were primiparous mothers and 43.2% were multiparous mothers (see Table 1).

Mothers who did participate in the study and those who did not participate differed significantly in the following sociodemographic variables: participants were older, primiparous, of Spanish nationality, they had a higher educational level, and higher economic income. However, there were no significant differences in terms of their marital status, or depending on whether or not they lived with their partner.

## Semantic equivalence

All items could be translated without problems into the Spanish cultural context. Most items (2, 4, 5, 6, 7, 8, 11, 13) did not require changes and retained the same syntactic structure in the Spanish version. A few syntactic or semantic changes were made to some items to obtain a semantically and conceptually equivalent scale appropriate to the Spanish spoken in Spain. For example, "look after" was changed to "take care of" ("cuidar" in Spanish) was used in Item 1. In addition, some different cultural expressions more appropriate to the Spanish context were used in some items, for example, "manage the situation" (English translation) was used instead of "cope with" in Item 3. Finally, in all items, the generic term "niño-a" was used to refer to the "baby/toddler "options of the original version. The average difficulty of translating the items, on a scale of 1 (*no difficulty*) to 10 (*maximum difficulty*), was 1.1.

**Table 1 Characteristics of the study sample ($n = 579$).**

| Variable | Mothers ($n = 579$) |
|---|---|
| **Age**, mean (SD) | 34.2 (4.2) |
| **Marital status** | |
| Married/ de facto | 405/509 (79.6) |
| Single | 89/509 (17.5) |
| Separated/ Divorced | 12/509 (2.3) |
| Widow | 3/509 (0.6) |
| **Cohabiting with partner** | |
| Yes | 489/508 (96.2) |
| Some days of the month | 10/508 (2.0) |
| No | 9/508 (1.8) |
| **Parity** | |
| Primiparous | 325/572 (56.8) |
| Multiparous | 247/572 (43.2) |
| **Educational status** | |
| Primary school or lower | 23/511 (4.5) |
| High School | 96/511 (18.8) |
| High School senior or vocational training | 109/511 (21.3) |
| University | 283/511 (55.4) |
| **Family annual incomes, in euros** | |
| Less than 6,000 | 30/492 (6.1) |
| 6,000–8,999 | 26/492 (5.3) |
| 9,000–11,999 | 51/492 (10.4) |
| 12,000–17,999 | 108/492 (21.9) |
| 18,000–29,999 | 142/492 (28.9) |
| 30,000–44,999 | 84/492 (17.1) |
| 45,000–60,000 | 33/492 (6.7) |
| More than 60,000 | 18/492 (3.6) |
| **Nationality** | |
| Spanish | 529/554 (95.5) |
| Other | 25/554 (4.5) |

**Notes.**
Data are presented as number and percentage (%) unless otherwise indicated.
SD, standard deviation.

## Descriptive findings of BaM-13 scale

The median score of the total scale was 7 (IQR = 4 −11.5), for a score ranging between 0 and 39. All items, except for Item 10 ("I have been worried that something would happen to my child"), presented positive skewness (see Table S1 ), and this item was the only one where 59.1% of mothers expressed concern about their child (see Table 2). Items 1 ("I have felt confident taking care of my child") and 4 ("I have felt close to my child") showed greater skewness compared to the other items, due to the floor effect they presented, concentrating more than 80% of the responses in category 0. The distribution of the items did not fit the normal distribution.

**Table 2  Item category frequencies (%).**

| Item | 0 = No, rarely or never | 1 = No, not very often | 2 = Yes, some of the time | 3 = Yes, most or all of the time |
|---|---|---|---|---|
| 1. I have (not) felt confident about taking care my child[*] | 85.1 | 14.2 | 0.5 | 0.2 |
| 2. I have missed the life I had before I became pregnant with this child | 45.9 | 26.6 | 27.5 | 0 |
| 3. I have found it hard to manage the situation when my child cries | 39.7 | 34.6 | 23.3 | 2.4 |
| 4. I have (not) felt close to my child[*] | 97.4 | 2.4 | 0.2 | 0 |
| 5. I have felt lonely or isolated | 61.5 | 18.3 | 19.5 | 0.7 |
| 6. I have felt bored | 67.9 | 21.9 | 10.2 | 0 |
| 7. I have felt unsupported | 60.3 | 18.1 | 20.6 | 1.0 |
| 8. I have (not) felt alright about asking people for help or advice when I needed to[*] | 61.8 | 32.8 | 5.0 | 0.4 |
| 9. I have felt nervous or uneasy around my child | 51.3 | 30.6 | 18.1 | 0 |
| 10. I have been worried that something would happen to my child | 15.2 | 25.7 | 54.1 | 5.0 |
| 11. I have been annoyed or irritated with my child | 56.5 | 26.6 | 16.7 | 0.2 |
| 12. I worry I am not as good as other mothers | 54.1 | 24.2 | 20.0 | 1.7 |
| 13. I have felt guilty | 58.2 | 23.1 | 18.1 | 0.6 |

**Notes.**
[*]Reversed direction of the question: 0 (Yes, most or all of the time), 1 (Yes, some of the time), 2 (No, not very often) and 3 (No, rarely or never).

## Construct validity

The Kaiser-Meyer-Olkin index was adequate (KMO $= 0.852$) and the Bartlett sphericity test was significant ($p < 0.001$). These findings indicated that the dataset was appropriate for performing factorial analysis. According to the results of the principal component analysis, three possible factors with eigenvalues greater than 1 (4.24, 1.35, and 1.02, respectively) were observed. The last-elbow rule on the screeplot of the eigenvalues suggested retaining two principal components (see Fig. 2). Finally, considering these two statistical criteria and taking into account the theoretical structure of the scale and the items, we decided to retain the two principal components that explained 43% of the total variability. As shown in Table 3, the factorial loads of the items were greater than 0.40, except for Item 4 (0.32). In the resulting structure, the items corresponding to Factors 1, Child experience, and 3, Emotional closeness, on the original scale were grouped into a single factor, called Postnatal bonding. Item 2 ("I have missed the life I had before I became pregnant with this child") loaded on both factors, with a greater weight on Factor 1 Postnatal bonding (0.52), than on Factor 2 Adult experience (0.31), its place on the original scale. We decided that Item 2 should remain in the factor where it had the highest loading, Factor 1.

Table 4 shows the goodness-of-fit indices obtained from the CFA of the three models tested: bifactorial, trifactorial, and unifactorial. The results indicated that the different models fit the data appropriately but the two-factor model showed better fit indicators. Therefore, taking into account the results of the exploratory factor analysis, as well as the

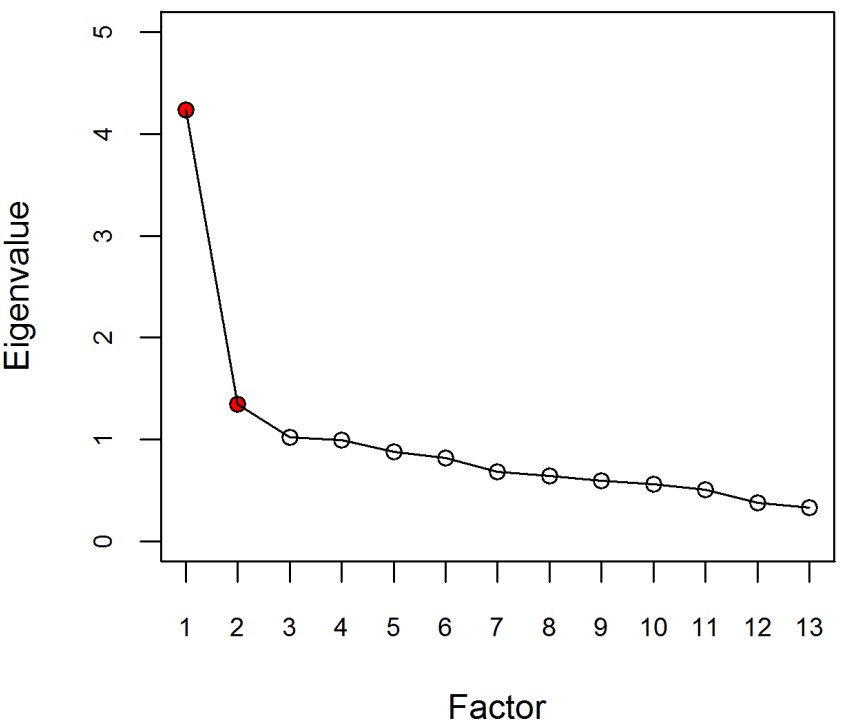

**Figure 2  Scree plot.**

goodness-of-fit indices, the two-factor model was selected to define the structure of the Spanish version of the BaM-13. Figure 3 shows a diagram of the final model.

### Internal consistency

The Cronbach alpha coefficients ($\alpha$) and omega coefficients ($\omega$) were adequate for the global scale ($\alpha = 0.818$, $\omega = 0.861$), and for Factors 1 ($\alpha = 0.773$, $\omega = 0.784$) and 2 ($\alpha = 0.710$, $\omega = 0.721$). The correlation coefficient between the factors was moderate ($r = 0.456$), indicating that the two factors were not redundant. The median scores were as follows: 7 for the total scale, 5 for Factor 1, and 1 for Factor 2 (see Table 5). The internal consistency of the scale showed a slight improvement by removing Item 4 ($\alpha = 0.779$) and 6 ($\alpha = 0.726$). The correlations of each item with its corresponding factor were greater than 0.40, except for Item 4 ($r = 0.227$) (see Table S1).

### Criterion validity

As detailed in Table 5, the three variables of interest (postpartum depression, parental competence, and postnatal bonding) were significantly associated with the experience of motherhood, with the coefficients being positive in the case of postpartum depression ($r = 0.560$) and negative in the case of parental competence ($r = -0.584$) and postnatal bonding ($r = -0.327$). As mentioned above, higher scores on the BaM-13 scale indicate a higher degree of dissatisfaction. Therefore, greater dissatisfaction with the experience of

**Table 3  Principal component analysis followed by varimax rotation (construct validity).**

| Item | Being a mother scale | |
|---|---|---|
| | RC1 Postnatal bonding | RC2 Adult's experience |
| 1. I have (not) felt confident about taking care my child (EC) | 0.55 | |
| 2. I have missed the life I had before I became pregnant with this child | 0.52 | |
| 3. I have found it hard to manage the situation when my child cries (CE) | 0.62 | |
| 4. I have (not) felt close to my child (EC) | 0.32 | |
| 5. I have felt lonely or isolated | | 0.75 |
| 6. I have felt bored | | 0.45 |
| 7. I have felt unsupported | | 0.84 |
| 8. I have (not) felt alright about asking people for help or advice when I needed to | | 0.70 |
| 9. I have felt nervous or uneasy around my child (CE) | 0.70 | |
| 10. I have been worried that something would happen to my child (CE) | 0.52 | |
| 11. I have been annoyed or irritated with my child (CE) | 0.62 | |
| 12. I worry I am not as good as other mothers (CE) | 0.65 | |
| 13. I have felt guilty (CE) | 0.65 | |

Notes.

EC, Emotional Closeness, corresponding to factor 3 on the original scale; CE, Child Experience, corresponding to factor 1 on the original scale.

**Table 4  Goodness-of-fit indices for the BaM-13.**

| Model | $\chi^2$ | df | $\chi^2$/df | CFI | TLI | RMSEA (90%CI) | SRMR |
|---|---|---|---|---|---|---|---|
| 1 factor | 181.846 | 65 | 2.798 | 0.960 | 0.952 | 0.056 (0.046–0.065) | 0.067 |
| 2 factors | 99.013 | 64 | 1.547 | 0.988 | 0.985 | 0.031 (0.018–0.042) | 0.050 |
| 3 factors | 115.913 | 62 | 1.870 | 0.982 | 0.977 | 0.039 (0.028–0.050) | 0.051 |

Notes.

$\chi^2$, chi square; df, degrees of freedom; CFI, Comparative fix index; TLI, Tucker Lewis index; RMSEA, Root Mean Square Error of Approximation; 90%CI, 90% confidence interval; SRMR, Standardized Root Mean Square Residual.

**Table 5  Intercorrelation, internal consistency and criterion validity.**

| Factor | Factor | | Median (IQR) | Cronbach $\alpha$ | Coefficient omega | Criterion validity | | |
|---|---|---|---|---|---|---|---|---|
| | Factor 1 | Factor 2 | | | | EPDS | PSOC | MPAS |
| Factor 1 | | | 5 (3–8.5) | 0.773 | 0.784 | 0.503[*] | −0.564[*] | −0.348[*] |
| Factor 2 | 0.456[*] | | 1 (0–3) | 0.710 | 0.721 | 0.472[*] | −0.424[*] | −0.166[*] |
| BaM-13 Total score | 0.934[*] | 0.727[*] | 7 (4-11.5) | 0.818 | 0.861 | 0.560[*] | −0.584[*] | −0.327[*] |

Notes.

*Statistically significant Spearman Correlation Coefficients at 0.05 significance level.

IQR, Interquartile range; EPDS, Edinburgh Postnatal Depression Scale; PSOC, Parenting Sense of Competence; MPAS, Maternal Postnatal Attachment Scale.

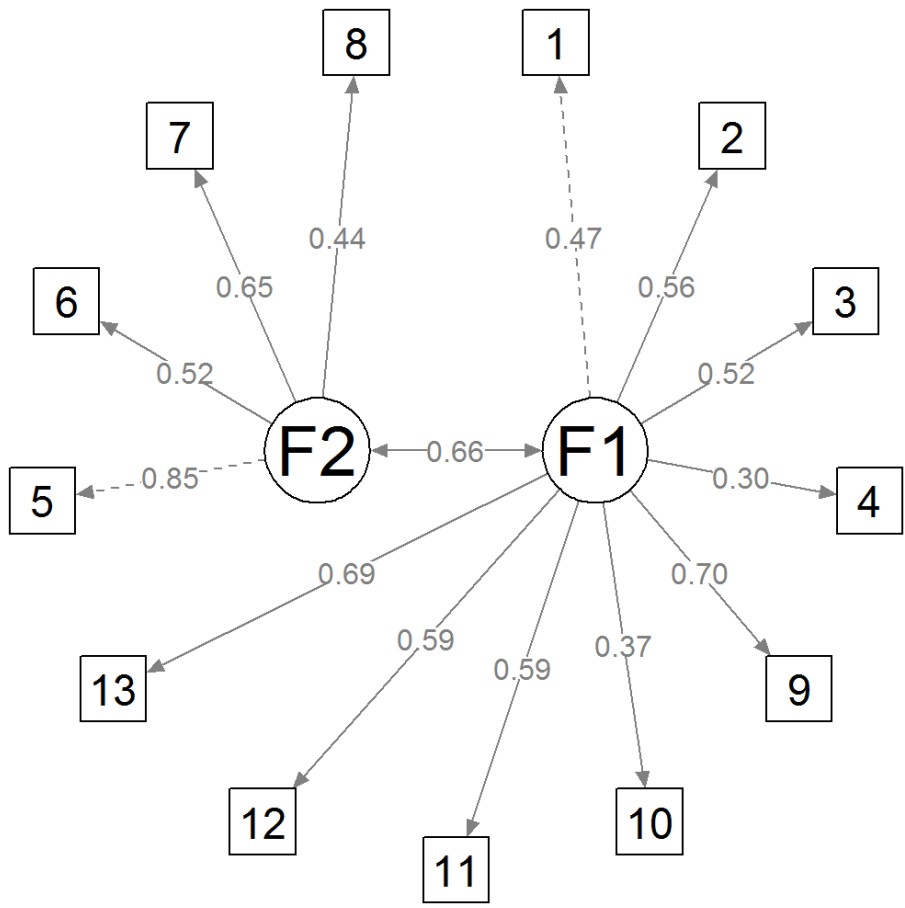

**Figure 3  Diagram of the final model.**

motherhood is associated with higher scores in postpartum depression, lower perceived parental competence, and lower postnatal bonding.

## Differences between primiparous and multiparous mothers

As detailed in Table 6, the primiparous mothers scored statistically higher in Items 1 ("I have felt confident taking care of my child"), 3 ("I have found it hard to manage the situation when my child cries"), 9 ("I have felt nervous or uneasy around my child"), 12 ("I worry I am not as good as other mothers"), and 13 ("I have felt guilty"); on the contrary, the multiparous mothers obtained a statistically higher score in Item 8 ("I have felt alright about asking people for help or advice when I needed to"). As mentioned above, Items 1 and 8 were reverse-drafted.

In terms of factors and total score, only Factor 1 Postnatal bonding was statistically significant, and higher in primiparous mothers. However, no significant differences were found in the overall maternity experience (global score), or in the mothers' experience as an adult (Factor 2).

**Table 6   Item endorsement and total BaM-13 score by parity.**

| Items & factors | Parity (n = 572) | | P-value |
| --- | --- | --- | --- |
| | Primip (n = 325) | Multip (n = 247) | |
| **Factor 1: Postnatal bonding** | 6 (3–9) | 5 (2–8) | 0.009 |
| 1. I have (not) felt confident about taking care my child | 0 (0–0) | 0 (0–0) | 0.012 |
| 2. I have missed the life I had before I became pregnant with this child | 1 (0–2) | 1 (0–2) | 0.280 |
| 3. I have found it hard to manage the situation when my child cries | 1 (0–2) | 1 (0–1) | 0.002 |
| 4. I have (not) felt close to my child | 0 (0–0) | 0 (0–0) | 0.798 |
| 9. I have felt nervous or uneasy around my child | 1 (0–1) | 0 (0–1) | 0.006 |
| 10. I have been worried that something would happen to my child | 2 (1–2) | 2 (1–2) | 0.394 |
| 11. I have been annoyed or irritated with my child | 0 (0–1) | 0 (0–1) | 0.613 |
| 12. I worry I am not as good as other mothers | 1 (0–1) | 0 (0–1) | 0.011 |
| 13. I have felt guilty | 0 (0–1) | 0 (0–1) | 0.052 |
| **Factor 2: Adult's experience** | 1 (0–3) | 1 (0–4) | 0.237 |
| 5. I have felt lonely or isolated | 0 (0–1) | 0 (0–1) | 0.676 |
| 6. I have felt bored | 0 (0–1) | 0 (0–1) | 0.548 |
| 7. I have felt unsupported | 0 (0–1) | 0 (0–2) | 0.244 |
| 8. I have (not) felt alright about asking people for help or advice when I needed to | 0 (0–1) | 0 (0–1) | 0.031 |
| **BaM-13 Total Score** | 7 (4–12) | 7 (4–11) | 0.143 |

Notes.
Data are presented as median (interquartile range).

# DISCUSSION

To our knowledge, this is the first study to report the application of the BaM-13 scale in Spain and to examine its psychometric properties in a large sample of mothers in the postpartum period. The Spanish version of the BaM-13 scale presents a two-factor model: Postnatal bonding, consisting of 9 items; and Adult's Experience, consisting of 4 items. The results show good evidence of the reliability and validity of this adaptation.

The translation of the BaM-13 scale followed a rigorous and systematic process to ensure the semantic equivalence with the original version. During the cognitive interviews, participants showed adequate interpretation and understanding of all questionnaire items, and none of them required special attention.

As for the behavior of the items, they showed an adequate distribution, except for Items 1 and 4, which had a floor effect. However, the factorial scores showed good performance. In this sense, when observing the percentage of responses with a score equal to 0 and the standard deviations (SD) of the factors and the total score of the scale (Factor 1 = 5.18%, 2.91 SD; Factor 2 = 31.61%, 2.17 SD; Total score = 3.80%, 5.33 SD), these scores can be considered to show no floor effect and to be sensitive to changes caused by circumstances that improve or worsen the mothers' experience.
Concerning structural validity, the Spanish version of the BaM-13 scale presented a satisfactory fit to a two-factor model. This structure was chosen with theoretical and empirical criteria. Concerning the theoretical criteria, we consider that the contents of the elements of Factor 1 Child Experience and Factor 3 Emotional Closeness of the original structure of the scale (*Matthey, 2011*) are grouped within the same concept, that of postnatal bond. This concept is defined as the parents' feelings toward their baby (*Taylor et al., 2005*; *Brockington, Aucamp & Fraser, 2006*; *Riera-Martín et al., 2018*), such as feelings of confidence or insecurity, longing, union or closeness, worry, anger, guilt, *etc*. These feelings are included in the different items of Factors 1 and 3 of the original scale. On the other hand, we have observed a strong parallelism between the items of these two factors with other scales that measure postnatal bond, for example, the Spanish version of Postpartum Bonding Questionnaire (PBQ) (*Garcia-Esteve et al., 2015*) and the PBS (*Riera-Martín et al., 2018*). For example, Item 11 of the BaM-13 scale ("I have been annoyed or irritated with my child"), would be equivalent to Item 10 of the PBQ ("I feel that my baby irritates me") and to Item 1 of the PBS ("When I am taking care of the baby, I feel annoyed or irritated"), among others. In addition, we observed a moderate and significant correlation between Factor 1 of the Spanish version of the BaM-13 scale and the PBS. Therefore, we consider that the concept of postnatal bond, one of the main constructs of the BAM theory, is the appropriate term for Factor 1 of the Spanish version of the scale, which unifies Factors 1 and 3 of the original scale.

The empirical criteria are as follows: first, the criteria used in exploratory factor analysis that determined the factors to be retained (eigenvalues greater than 1 and last-elbow rule) suggested this structure; second, better goodness-of-fit indices compared to the other two models (one factor, three factors); and finally, the appropriate level of internal consistency subsequently evaluated. The only notable aspect is the factorial loads of Items 4 and 2. Regarding Item 4, its lower factorial load could be explained because of the floor effect: option 0 was selected in 97.4% of the responses given. On the other hand, Item 2 loads on the two factors, but higher on Factor 1. Conceptually, it makes sense for it to load on both factors, because it compares life before having a child (adult experience) with the current life with the child (postnatal bond). As a result, we decided to retain Item 2 in Factor 1.

The internal consistency of the global scale was adequate, obtaining a value similar to that found in the original version of the scale (*Matthey, 2011*), the American version (*Henshaw et al., 2015*) and the German version (*Göbel et al., 2020*). Likewise, the internal consistency of both factors was adequate, but this information could only be contrasted with the version adapted to German (*Göbel et al., 2020*) because in the original version (*Matthey, 2011*), as in the other adaptations mentioned (*Henshaw et al., 2015*; *Pontoppidan, 2015*), these data were not presented. In this sense, the internal consistency of the subscales in the German version presented difficulties in the Emotional Closeness factor ($\alpha = 0.10$), so the authors decided not to include the factor in subsequent analyses.

Concerning criterion validity, the scale showed statistically significant correlations in the expected direction with postpartum depression, parental competence, and postnatal bonding. This implies that mothers who have difficulties in the process of becoming a mother also have greater depressive symptomatology (*Gavin et al., 2005*), less perceived

parental competence (*Chapman & Hart, 2017*), and a weaker postnatal bond (*De Cock et al., 2016*). Additionally, these results showed that although there is a reasonable level of concordance between the different scales and the Bam-13, they measure different constructs (as the *r* does not obtain a value around 0.8 or higher). This is especially interesting as it shows the difference between the construct assessed by the BaM-13 and depressive symptoms in the postpartum period. Therefore, this tool does not assess postpartum depression.

As for the impact of parity on the experience of motherhood, our findings present some heterogeneity, in agreement with the variability found in the scientific literature. On the one hand, we found significant differences between the two groups of mothers, which deserves special attention. Primiparous mothers have greater difficulty in postnatal bonding (Factor 1), compared to multiparous mothers. These results are in line with various studies (*Göbel et al., 2020*; *Nakano et al., 2019*; *Rossen et al., 2017*; *Tsuchida et al., 2019*; *Yoshida et al., 2020*). Like other previous articles, our hypothesis indicates lower confidence in baby care or a lower perception of parental competence (*Coleman & Karraker, 1998*; *Kim et al., 2013*; *Shorey et al., 2014*). As mentioned in the study by *Göbel et al. (2020)*, another explanation could be higher pregnancy-related anxiety and hostility, which could be greater in primiparous women.

On the other hand, parity did not play an explanatory role in the overall score of the experience of motherhood, or the mother's experience as an adult (Factor 2). These results are in line with the studies of various authors (*Bassi et al., 2017*; *Osnes et al., 2021*) who also found no significant differences between primiparous and multiparous mothers. These findings indicate that multiparity does not have a protective effect on the difficulties that may arise in the experience of motherhood, as opposed to the traditional perspective, which assumes that the multiparous mother has acquired the maternal role and it attributes fewer difficulties and less need for support to her (*Gameiro, Moura-Ramos & Canavarro, 2009*). Therefore, we consider multiparity as a process with its own particularities, such as a more complex family reorganization, adapting to the baby's arrival while continuing to care for her other children who are at different stages of development, as well as increasing the demands of time and energy (*Krieg, 2007*).

## Limitations and suggestions for future research

There are some limitations to this study. First, the data were obtained through self-reported questionnaires, with possible bias in the responses obtained, such as social desirability. Secondly, the homogeneity of the sample characteristics does not allow generalizing the results. It would be desirable to extend the study to more heterogeneous samples in terms of the mothers' age, socioeconomic level, educational level, nationality, and associated clinical problems, among others. Third, the design of the study is cross-sectional, so there are no data on how the experience of becoming a mother evolves in the first months and years after the baby is born. Finally, in this study, test-retest reliability and measurement error were not measured.

Four lines of future research are recommended. Firstly, we consider it of interest to study the influence of dyadic relationships (between parents) in the experience of becoming a

mother. Secondly, we recommend studying the influence of family contextual factors, such as parental stress and the quality of the couple relationship. Thirdly, studies should be carried out to allow the adaptation of this instrument to measure the experience of paternity. Finally, we consider it of interest to extend the study to the new existing family structures: families with premature children, reconstituted families, single-parent families, gay couples, adoption, surrogate motherhood-fatherhood, and immigration processes.

## CONCLUSIONS

The Spanish version of the BaM-13 scale is valid for measuring mothers' experience of motherhood in a wide range of domains. This tool will provide professionals in the field of perinatal mental health to identify and address the emotional difficulties that mothers may experience in the transition to motherhood. On the other hand, the findings of the study suggest considering parity in the care of women during pregnancy and postpartum, highlighting the importance of addressing the postnatal bond in primiparous mothers. Additionally, we highlight the need not to assume that multiparous mothers experience fewer difficulties in their maternity and to offer the necessary attention to the complex process they undergo.

## ACKNOWLEDGEMENTS

The authors thank Alejandro Cerezo Munuera for his collaboration in the translating process of the scale.

### Funding

This project was supported by the General Sub-Directorate for Evaluation and Promotion of Research (Institute of Health Carlos III, ISCIII) and co-funded by the European Regional Development Fund (FEDER) (No. PI14/01549). The funders had no role in study design, data collection and analysis, decision to publish, or preparation of the manuscript.

### Grant Disclosures

The following grant information was disclosed by the authors:
General Sub-Directorate for Evaluation and Promotion of Research.
European Regional Development Fund (FEDER): PI14/01549.

### Competing Interests

The authors declare there are no competing interests.

### Author Contributions

- Anna Riera-Martín conceived and designed the experiments, performed the experiments, analyzed the data, prepared figures and/or tables, authored or reviewed drafts of the article, and approved the final draft.

- Antonio Oliver-Roig conceived and designed the experiments, performed the experiments, analyzed the data, prepared figures and/or tables, authored or reviewed drafts of the article, and approved the final draft.
- Susana Cormenzana conceived and designed the experiments, performed the experiments, analyzed the data, prepared figures and/or tables, authored or reviewed drafts of the article, and approved the final draft.
- Miguel Richart-Martínez conceived and designed the experiments, performed the experiments, analyzed the data, prepared figures and/or tables, authored or reviewed drafts of the article, and approved the final draft.
- Ana Martínez-Pampliega conceived and designed the experiments, performed the experiments, analyzed the data, prepared figures and/or tables, authored or reviewed drafts of the article, and approved the final draft.

## Human Ethics

The following information was supplied relating to ethical approvals (i.e., approving body and any reference numbers):

The study was approved by the Ethical Committee of Clinical Research of the General Direction of Public Health and Higher Center of Research in Public Health (CEIC-DSGSP/CSISP), attached to the Health Council of the Valencian Community.

## Data Availability

The raw data are provided in the Supplementary Files.

## Supplemental Information

Supplemental information for this article can be found online at http://dx.doi.org/10.7717/peerj.18015#supplemental-information.

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
