# Peer review of "Adaptation and validation of the Spanish version of the Being a Mother scale"

_PeerJ, doi:10.7717/peerj.18015_

## Round 0.1 · original submission · Minor Revisions

I have received two thoughtful and extremely constructive reviews of your article. Both view the article and the study positively, but also identify a number of places where the article could be improved. While the number of comments is large, I am issuing a decision of "minor revisions" since most/all relate to adding information or clarification to the text (rather than, for example, requiring additional data collection or extensive new analyses). Please give consideration to all these comments as you prepare a revision.

·

Basic reporting

I suggest the authors the following:
1. To reference the translation process in the title, e.g., Translation a validation of the... or Translation and psychometric properties of the ...
2. To use along the manuscript the term European-Spanish, as it was culturally adapted to mothers from Spain (and not Latino American countries).
3. To rephrase the last sentence of results in the abstract. "Parity" is not mentioned before in the abstract, and non-specialized readers can find it out of context. Consider including a sentence structure like "Multiparous/Primiparous mothers showed... than primiparous/multiparous mothers. Please also interpret the result in Postnatal Bounding, instead of just mentioning significant differences.
4. The last sentence of the conclusions in the abstract
5. In line 149-151, please provide what items corresponded to each factor in the original scale.
6. Please, review and be consistent with the use of postpartum depression or postnatal depression.
7. Please, provide how Spanish proeficiency was evaluated in participants with a non-Spanish nationality.
7. Line 178. Provide a reference supporting the translation process.
8. I think it would be helpful if an appendix provided the back-translated versions of their adapted Bam-13 for readers who are not fluent in Spanish.
9. Line 182, What are cognitive interviews? Are they intended for content/face-to-face validity? More details are needed about his process.
10. Line 253, the paragraph presents no information described in Methods (e.g., the scale of difficulty), and unclear information is provided from "cognitive interviews" with 10 mothers.
11. Line 264. It would be helpful to provide an appendix with histograms for the total scores (and factors) for the whole sample and divided by subgroups. This will be helpful to discuss the presence of floor/ceiling effects in the summed scores (not only item by item).
12. Line 275, Only criteria for excellent (and not for adequate after KMO) was explained in Methods.
13. Line 310, when there is a large sample size, is not informative enough presenting only significative correlations, and magnitude should also be presented.
14. The discussion section is hard to follow, and including some subheadings (like for limitations) can help the reader.
15. Line 415. "Dyadic relationship" between who?
16. Table 6, Please group items by factor for an easy understanding of the table.

Experimental design

One limitation that the authors should acknowledge is that due to becoming/being a mother is a process, the sample recruited in the study, concerning only mothers with a newborn between 6-11 months, cannot reflect this process.
In this context, I would like to ask the authors to justify in the manuscript the deviations from the eligibility criteria in your study compared to the original validation of the scale (Matthey, 2011), and discuss the implications in your results and its interpretation.

Please, provide a scientific-based rationale and a priori hypothesis of criterion validity for the PSOC and MPAS tools.

In line 206-207, you mentioned that the scale was evaluated according to the COSMIN checklist. However, the use of terms regarding internal consistency and reliability is not aligned with this checklist. Cronbach Alpha is intended to assess internal consistency. Reliability was not measured in this study, as no test-retest was conducted. This aligns with the fisrt limitation mentioned: being a mother is a process. Not presenting test-retest reliability values, as well as the error of measurement of the tool, is a limitation also necessary to be acknowledged.

Validity of the findings

Please, consider rephrasing the last sentence of the conclusions in the abstract, as the current statement cannot be concluded from the results of this study.

Regarding Figure 1, can you provide whether the included mothers in the study were representative of the whole screening mother complying with the eligibility criteria?

Please justify why you chose to conduct an EFA instead of a CFA with the 3 preindentified factors. EFA implies that researchers a priori expect to find a different structure than in the original scale, but this rationale was not presented.
Even if the goodness-of-fit indexes were slightly better for the 2 factors solution, this could not be the sole argument for its final selection as, for example, providing a scale (if valid) with a similar structure to the original will allow the comparison of multicountry studies.

The rationale in Methos to maintain an item was presenting factor loadings >0.4. However, both the loading factor of 0.32 for item 4, as well as its low correlations presented in line 306, indicated that this item should be removed from the final version. Item 4 needs to be discussed, and the potential reassons behind its bad fit must be argued.

Line 331. Is there not a version of the BaM-13 for Spanish speakers in other countries?

Line 355-365. You adequately describe why F1 and F3 of the original scale fit in the Spanish version of the BaM-13, but also, justification about why the F2 (Child's experience) was not appropriate is needed.

Line 419. Presenting the different family structures, how did you ensure that this model of families was not included in the current study (e.g., single-parent families)?

Already been mentioned, but it is necessary to review the use of the term reliability vs internal consistency according to COSMIN.

Additional comments

Congratulations to the authors for the work done and presented.

Reviewer 2 ·

Basic reporting

The manuscript is well written, with professional English easy to follow.
Literature background is well presented.

I have one minor revision regarding the label "coexistence with partner", as this might not be such a common term. Cohabiting might be the better fitting term in this context

It would be interesting to add some information on the independence of the construct assessed with BaM-13 compared to symptoms of depression (as argued by Matthey, 2011), as one could argue that the experience assessed with the BaM-13 partly correspond with depressive symptoms in the postpartum period. Your results reported later on associations with BaM-13 and EPDS scores could then be discussed in this context.

Experimental design

The experimental design and aims of the study are well described, methods are easy to follow. investigation is rigorous.

It would be beneficial to add some statistical background information, regarding:
- Information on statistical power analysis
- Information on dealing with missings
- Information on why you decided to compare scores between primi- and multiparous women using nonparametric Wilcoxon test
- Some explaination why you decided to report both Cronbach's alpha and McDonald's omega

- I think it would be really interesting and add deeper understanding in the context of criterion validity to report correlations with PSOC separately for the satisfaction and the self-efficacy domain

- I was wondering whether it would be interesting to present next to the total score associations of BaM-13 also the separate results for the EPDS sadness and anxiety subscales, too, in case there is already convincing data on their validty in the Spanish EPDS. This point is really more a suggestion and open question to you.

Regarding the presentation of the results, minor changes to the manuscript might be beneficial:

- Please add some information on how far you did interpret Chi-square, as some reseachers suggest that in samples >300 this value might not be valid for reporting model fit

- In Table 3 there is only one item marked in Italics; in general, as you extract a new factor combining items from two original subscales and the scale has a new label, this scale does not represent the original scale. Assignment to the original scales might be easier to understand if you add this information behind the item or in a separate column, for example:
“have (not) felt confident about taking care of my child (EC)” with EC for emotional closeness

Validity of the findings

The findings are very informative and interesting to researchers in this field, as there is only a few studies investigating the psychometric properties and factorial validity of the BaM-13. The results can be of great support for researchers aiming to investigate the transition to motherhood.

The discussion is linked to the original research and adds informative perspectives to the research field.

Limitations are well discussed.

It might be interesting to add information from Göbel et al. 2020 on the relevance of parity on the BaM-13 scale (as they report details on associations on BaM-13 subscale level that contribute to your discussion of the results); https://doi.org/10.1016/j.midw.2020.102824

**Staff Note: It is PeerJ policy that additional references suggested during the peer-review process should only be included if the authors are in agreement that they are relevant and useful**

Additional comments

This is a really interesting study and relevant for promoting research in this area.

---

## Round 0.2 · accepted · Accept

I agree with the reviewer that you have made thoughtful and complete responses to the reviewer comments, and that the manuscript is ready for publication.

Congratulations on a nice study and a well-written article!

·

Basic reporting

'no comment'

Experimental design

'no comment'

Validity of the findings

'no comment'

Additional comments

I congratulate the authors for their approach to the comments. I have no further comments.